# Appetite loss at discharge from acute decompensated heart failure: Observation from KCHF registry

**Erika Yamamoto**[1], **Takao Kato**[1]*, **Hidenori Yaku**[2], **Takeshi Morimoto**[3], **Yasutaka Inuzuka**[4], **Yodo Tamaki**[5], **Neiko Ozasa**[1], **Yusuke Yoshikawa**[1], **Takeshi Kitai**[6], **Ryoji Taniguchi**[7], **Moritake Iguchi**[8], **Masashi Kato**[2], **Mamoru Takahashi**[9], **Toshikazu Jinnai**[10], **Tomoyuki Ikeda**[11], **Kazuya Nagao**[12], **Takafumi Kawai**[13], **Akihiro Komasa**[1], **Ryusuke Nishikawa**[14], **Yuichi Kawase**[15], **Takashi Morinaga**[16], **Mitsunori Kawato**[17], **Yuta Seko**[1], **Masayuki Shiba**[1], **Mamoru Toyofuku**[18], **Yutaka Furukawa**[19], **Yoshihisa Nakagawa**[20], **Kenji Ando**[16], **Kazushige Kadota**[15], **Satoshi Shizuta**[1], **Koh Ono**[1], **Yukihito Sato**[7], **Koichiro Kuwahara**[21], **Takeshi Kimura**[1]

1 Department of Cardiovascular Medicine, Kyoto University Graduate School of Medicine, Kyoto, Japan, 2 Department of Cardiology, Mitsubishi Kyoto Hospital, Kyoto, Japan, 3 Department of Clinical Epidemiology, Hyogo College of Medicine, Hyogo, Japan, 4 Department of Cardiovascular Medicine, Shiga General Hospital, Shiga, Japan, 5 Division of Cardiology, Tenri Hospital, Nara, Japan, 6 Department of Cardiovascular Medicine, National Cerebral and Cardiovascular Center, Osaka, Japan, 7 Department of Cardiology, Hyogo Prefectural Amagasaki General Medical Center, Hyogo, Japan, 8 Department of Cardiology, National Hospital Organization Kyoto Medical Center, Kyoto, Japan, 9 Department of Cardiology, Shimabara Hospital, Kyoto, Japan, 10 Department of Cardiology, Japanese Red Cross Otsu Hospital, Shiga, Japan, 11 Department of Cardiology, Hikone Municipal Hospital, Shiga, Japan, 12 Department of Cardiology, Osaka Red Cross Hospital, Osaka, Japan, 13 Department of Cardiology, Kishiwada City Hospital, Osaka, Japan, 14 Department of Cardiology, Shizuoka General Hospital, Shizuoka, Japan, 15 Department of Cardiology, Kurashiki Central Hospital, Okayama, Japan, 16 Department of Cardiology, Kokura Memorial Hospital, Fukuoka, Japan, 17 Department of Cardiology, Kobe City Nishi-Kobe Medical Center, Hyogo, Japan, 18 Department of Cardiology, Japanese Red Cross Wakayama Medical Center, Wakayama, Japan, 19 Department of Cardiovascular Medicine, Kobe City Medical Center General Hospital, Hyogo, Japan, 20 Department of Cardiovascular Medicine, Shiga University of Medical Science, Shizuoka, Japan, 21 Department of Cardiovascular Medicine, Shinshu University Graduate School of Medicine, Matsumoto, Japan

* tkato75@kuhp.kyoto-u.ac.jp

## Abstract

### Objective

The complex link between nutritional status, protein and lipid synthesis, and immunity plays an important prognostic role in patients with heart failure. However, the association between appetite loss at discharge and long-term outcome remains unclear.

### Methods

The Kyoto Congestive Heart Failure registry is a prospective cohort study that enrolled consecutive patients hospitalized for acute decompensated heart failure (ADHF) in Japan. We assessed 3528 patients alive at discharge, and for whom appetite and follow-up data were available. We compared one-year clinical outcomes in patients with and without appetite loss at discharge.

**Data Availability Statement:** Data are available from the institutional review boards of Kyoto University (contact ethcom@kuhp.kyoto-u.ac.jp.) for researchers who meet the criteria for access to

confidential data. After ethical consideration in institutional review boards of Kyoto University, sharing a de-identified data set will be allowed. Based on the ACT on the Protection of personal Information, data sharing except in Japan will be allowed after the patient consent and ethical consideration.

**Funding:** This work was supported by the Japan Agency for Medical Research and Development [18059186] (Drs T. Kato, T. Kuwahara, and N. Ozasa). The founder had no role in the study design, collection, analysis or interpretation of the data, writing the manuscript, or the decision to submit the paper for publication.

**Competing interests:** The authors have declared that no competing interests exist.

## Results

In the multivariable logistic regression analysis using 19 clinical and laboratory factors with P value < 0.1 by univariate analysis, BMI < 22 kg/m$^2$ (odds ratio (OR): 1.57, 95% confidence interval (CI): 1.11–2.24, P = 0.01), CRP >1.0mg/dL (OR: 1.49, 95%CI: 1.04–2.14, P = 0.03), and presence of edema at discharge (OR: 4.30, 95%CI: 2.99–6.22, P<0.001) were associated with an increased risk of appetite loss at discharge, whereas ambulatory status (OR: 0.57, 95%CI: 0.39–0.83, P = 0.004) and the use of ACE-I/ARB (OR: 0.70, 95% CI: 0.50–0.98, P = 0.04) were related to a decreased risk in the presence of appetite loss. The cumulative 1-year incidence of all-cause death (primary outcome measure) was significantly higher in patients with appetite loss than in those without appetite loss (31.0% vs. 15.0%, P<0.001). The excess adjusted risk of appetite loss relative to no appetite loss remained significant for all-cause death (hazard ratio (HR): 1.63, 95%CI: 1.29–2.07, P<0.001).

## Conclusions

Loss of appetite at discharge was associated with worse 1-year mortality in patients with ADHF. Appetite is a simple, reliable, and useful subjective marker for risk stratification of patients with ADHF.

## Introduction

The complex link between nutritional status, protein and lipid synthesis, and immunity plays an important prognostic role in patients with heart failure (HF), although its role in acute decompensated HF (ADHF) remains to be elucidated. Several studies have reported an association between nutritional status and prognosis of heart failure patients [1–3]. Recently, several scoring systems have been reported to be associated with the prognosis of heart failure patients. The Prognostic Nutritional Index (PNI), which was originally developed for the prediction of postoperative complications, mainly in surgical patients, was associated with mortality in patients with heart failure [4]. Geriatric Nutritional Risk Index was also reported to be associated with prognosis in elderly heart failure patients [5]. Appetite is an important factor in improving the nutritional status of patients with heart failure.

Appetite is a desire for food and can be influenced by physical conditions such as blood sugar levels, hormones, and exercise. It can also be driven by mood and emotions and is assessed by healthcare providers in history-taking. In patients with ADHF, several residual symptoms at discharge such as pulmonary congestion, pitting edema, and pleural effusion were reportedly associated with long-term outcomes [6–9]. Although in the past years there were at least few important papers published concerning the issue of appetite loss and anorexia/cachexia syndrome in heart failure[3,10–13], there is still scarcity in the studies focused on "appetite loss," a subjective symptom in patients with heart failure [10,13]. This study aimed to clarify the characteristics of patients with appetite loss at discharge from ADHF and investigate the association between appetite loss at discharge and clinical outcomes at 1-year in patients with ADHF in Japan.

## Methods

### Study population

The Kyoto Congestive Heart Failure (KCHF) registry is a physician-initiated, prospective, observational, multicenter cohort study enrolling consecutive patients who were admitted

because of ADHF for the first time between October 2014 and March 2016 in 19 secondary and tertiary hospitals, including rural and urban, and large and small hospitals in Japan. The details of the KCHF registry have been described previously [14]. The KCHF registry enrolled all patients with acute decompensated heart failure defined by the modified Framingham criteria who were admitted to the participating centers and underwent heart failure-specific treatment involving intravenous drugs within 24 hours after hospital presentation. The study protocol was approved by the institutional review board of each participating center. Patient records were anonymized prior to the analysis. Written informed consent was waived by the institutional review boards of Kyoto University and each participating center, as the study met the conditions outlined in the Japanese ethical guidelines for medical and health research involving human subjects [15]. This study is reported in accordance with the Strengthening the Reporting of Observational Studies in Epidemiology (STROBE) reporting guidelines.

## Ethics

The study conformed to the principles outlined in the Declaration of Helsinki. The study protocol was approved by the ethics committees of each participating hospital. The approval number is provided in the supplementary appendix. Patient records were anonymized prior to the analysis. As the study met the conditions of the Japanese Ethical Guidelines for Medical and Health Research Involving Human Subjects [16], the institutional review boards of Kyoto University and each participating center waived the need to obtain written informed consent from each patient. We disclosed the details of the present study to the public as an opt-out method and clearly informed patients of their right to refuse enrollment. The details have been previously described.[10]

## Definitions of appetite loss and other variables

In KCHF study, symptoms and physical findings were evaluated using a four-level symptomatic grading (0, None; 1, Seldom/Mild; 2, Frequent/Moderate; 3, Continuous/Severe) at discharge [17]. As for appetite loss, attending physician asked the patient about the appetite ("Do you have an appetite?" with the response option "Yes/No". If patients answer was "No," the next question was "How bad is your appetite loss?" with the response option "1, Mild; 2, Moderate; 3, Severe." "Appetite loss" was defined the presence of any grade of appetite loss. "Mild appetite loss" was defined as the presence of appetite loss of grade 1, and "moderate/severe appetite loss" was defined as the presence of grade 2 or 3. Residual edema was defined as follows; 0, no edema; 1, slight pitting edema that disappeared rapidly; 2, moderate pitting edema that may last more than 1 min; and 3, severe pitting edema that lasted more than several minutes. Patient's ADL level was assessed on four level: (1) ambulatory (patients who can walk by themselves), (2) use wheelchair (outdoor only), (3) use wheelchair (outdoor and indoor), (4) bedridden. Other definitions for the baseline factors are provided in S1 File.

## Outcomes

Clinical follow-up data were collected in October 2017. The attending physicians or research assistants at each participating hospital collected clinical events after the index hospitalization from the hospital charts or by contacting patients, their relatives, or their referring physicians with consent. Detailed definitions of clinical outcome measures have been described previously [18]. The primary outcome measure in the current analysis was all-cause death. The secondary outcome measures were cardiovascular (CV) death, non-CV death, and HF hospitalization. The causes of death were classified according to the Valve Academic Research

Consortium definitions [19] and adjudicated by a clinical event committee (supplementary methods).

## Patient and public involvement

This study was conducted without patient involvement. Patients were not invited to comment on the study design and were not consulted to develop patient-relevant outcomes or interpret the results. The patients were not invited to contribute to the writing or editing of this document for readability or accuracy.

## Statistical analysis

Categorical variables are presented as counts with percentages and compared using the $\chi^2$ test. Continuous variables were expressed as the mean with standard deviation or median with interquartile range (IQR) and compared using the Student's $t$-test or the Wilcoxon rank-sum test, as appropriate. To determine the factors associated with appetite loss at discharge, we compared the clinical characteristics and the presence or absence of appetite loss at discharge. A multivariable logistic regression model was developed to identify clinical characteristics, laboratory data, and medications at discharge that were independently associated with the presence of appetite loss using 19 factors with P value < 0.1 by univariate analysis extracted from the variables listed in Table 1. The results are expressed as the odds ratios (ORs) with their 95% confidence intervals (CIs). The Kaplan–Meier method was used to estimate the cumulative 1-year incidence of clinical events, and differences were assessed using the log-rank test. The date of discharge from the index hospitalization was regarded as time zero for the clinical follow-up. We used a multivariable Cox proportional hazard model to estimate the hazard ratios (HRs) and their 95% CIs of patients with appetite loss relative to those without appetite loss for the primary and secondary outcome measures. Proportional hazard assumptions were assessed on the plots of log (time) versus log [− log (survival)] stratified by the variables and were verified to be acceptable. Consistent with the previous reports [20–24], we selected the 30 clinically relevant risk-adjusting variables listed in Table 1. For the risk-adjusting variables, continuous variables were dichotomized by clinically meaningful reference values or median values: age ≥80 years based on the median value; left ventricular ejection fraction (LVEF) <40% based on the HF guideline of LVEF classification [25]; body mass index (BMI) <22 kg/m²; estimated glomerular filtration rate <30 mL/min/1.73m²; decreased albumin levels (serum level <3.0 g/ dL); and hyponatremia (serum sodium level <135 mEq/L). As an exploratory analysis, we compared the 1-year clinical outcomes of patients with moderate/severe appetite loss, mild appetite loss, and no appetite loss. We also performed subgroup analyses stratified by age, renal function, activity of daily living (ADL), C-reactive protein (CRP) level, and type and number of medications. All statistical analyses were performed by a physician (E.Y.) and a statistician (T.M.) using JMP 13.0 (SAS Institute Inc., Cary, NC, USA). Statistical significance was set at P < 0.05.

## Results

### Study population

Among the 4056 patients enrolled in the KCHF registry, 271 died during the index hospitalization. Among the 3758 patients who were discharged alive, we excluded 200 patients because of missing appetite data. We also excluded 57 patients due to missing follow-up data after discharge. Finally, the study population for the present analysis consisted of 3528 patients who were discharged alive with data on appetite and follow-up data. There were 405 patients

**Table 1. Baseline patient characteristics.**

| Variables | Appetite loss (N = 405, 11.5%) | No appetite loss (N = 3123, 88.5%) | P value | N of patients analyzed |
|---|---|---|---|---|
| **Demographics** | | | | |
| Age, years*† | 83 [75–88] | 80 [71–86] | <0.001 | 3528 |
| Age ≥80 years | 257 (63.5) | 1568 (50.2) | <0.001 | 3528 |
| Men*† | 192 (47.4) | 1761 (56.4) | <0.001 | 3528 |
| BMI, kg/m² | 21.9 ± 4.2 | 23.0 ± 4.5 | <0.001 | 3361 |
| BMI <22 kg/m²*† | 214 (57.2) | 1337 (44.8) | <0.001 | 3361 |
| **Etiology** | | | <0.001 | 3528 |
| Coronary artery disease | 118 (29.1) | 1018 (32.6) | | |
| Acute coronary syndrome† | 26 (6.4) | 163 (5.2) | | |
| Cardiomyopathy | 60 (14.8) | 477 (15.3) | | |
| Valvular heart disease | 95 (23.5) | 594 (19.0) | | |
| Hypertensive heart disease | 82 (20.3) | 804 (25.7) | | |
| Other heart disease | 50 (12.4) | 230 (7.4) | | |
| **Medical history** | | | | |
| Prior hospitalization due to HF*† | 177 (45.4) | 1074 (34.9) | <0.001 | 3471 |
| Atrial fibrillation or flutter† | 178 (44.0) | 1317 (42.2) | 0.5 | 3528 |
| Hypertension*† | 267 (65.9) | 2290 (73.3) | 0.002 | 3528 |
| Diabetes mellitus*† | 129 (31.9) | 1183 (37.9) | 0.018 | 3528 |
| Dyslipidemia | 147 (36.3) | 1223 (39.2) | 0.27 | 3528 |
| Prior myocardial infarction† | 88 (21.7) | 708 (22.7) | 0.67 | 3528 |
| Prior stroke† | 68 (16.8) | 488 (15.6) | 0.55 | 3528 |
| Current smoking*† | 29 (7.4) | 401 (13.0) | 0.001 | 3480 |
| Chronic kidney disease | 202 (50.0) | 1348 (43.2) | 0.01 | 3528 |
| Chronic lung disease† | 44 (10.9) | 425 (13.6) | 0.13 | 3528 |
| Malignancy | 69 (17.0) | 444 (14.2) | 0.13 | 3528 |
| Dementia* | 107 (26.4) | 504 (16.1) | <0.001 | 3528 |
| **Social backgrounds and activities** | | | | |
| Living alone† | 77 (19.0) | 676 (21.7) | 0.22 | 3528 |
| Ambulatory*† | 260 (65.0) | 2533 (81.9) | <0.001 | 3493 |
| **Vital signs at presentation** | | | | |
| Systolic blood pressure, mmHg | 144 ± 35 | 148 ± 35 | 0.004 | 3521 |
| Systolic blood pressure <90 mmHg† | 16 (4.0) | 71 (2.3) | 0.04 | 3521 |
| Heart rate, bpm | 94 ± 27 | 96 ± 28 | 0.15 | 3507 |
| Heart rate <60 bpm† | 27 (6.7) | 215 (6.9) | 1.00 | 3507 |
| Body temperature >37.5 degree Celsius | 34 (9.1) | 172 (5.8) | 0.01 | 3365 |
| NYHA Class III or IV | 356 (87.9) | 2708 (87.1) | 0.65 | 3514 |
| **Tests at admission** | | | | |
| LVEF | 47 ± 17 | 46 ± 16 | 0.8 | 3517 |
| HFrEF (LVEF <40%)† | 142 (35.2) | 1152 (37.0) | | |
| HFmrEF (LVEF 40–49%) | 73 (18.1) | 593 (19.1) | | |
| HFpEF (LVEF ≥50%) | 189 (46.8) | 1368 (43.9) | | |
| BNP, pg/mL | 828 [431–1545] | 700 [389–1216] | <0.001 | 3112 |
| NT-proBNP, pg/mL | 9080 [3842–17211] | 5312 [2535–11457] | 0.03 | 641 |
| Creatinine, mg/dL | 1.21 [0.85–1.81] | 1.09 [0.82–1.57] | 0.08 | 3522 |
| eGFR, mL/min/1.73m² | 38.1 [24.1–57.5] | 45.3 [30.1–61.2] | <0.001 | 3522 |

(*Continued*)

**Table 1.** (Continued)

| Variables | Appetite loss | No appetite loss | P value | N of patients analyzed |
|---|---|---|---|---|
| | (N = 405, 11.5%) | (N = 3123, 88.5%) | | |
| eGFR <30 mL/min/1.73m$^2$† | 138 (34.2) | 776 (24.9) | <0.001 | 3522 |
| Blood urea nitrogen, mg/dL | 26 [18–38] | 23 [17–33] | <0.001 | 3518 |
| Serum sodium, mEq/L | 139 ± 5 | 139 ± 4 | 0.12 | 3515 |
| Sodium <135 mEq/L† | 60 (14.9) | 348 (11.2) | 0.03 | 3515 |
| Hemoglobin, g/dL | 11.2 ± 2.2 | 11.6 ± 2.4 | <0.001 | 3523 |
| Anemia†† | 292 (72.1) | 2033 (65.2) | 0.006 | 3523 |
| Albumin, g/dL | 3.4 ± 0.5 | 3.5 ± 0.5 | <0.001 | 3421 |
| Albumin <3.0 g/dL | 74 (19.1) | 377 (12.4) | 0.003 | 3421 |
| C reactive protein, mg/dL | 0.73 [0.20–3.09] | 0.58 [0.20–1.85] | <0.001 | 3206 |
| **Medication at discharge** | | | | |
| Number of prescribed drugs | 8 [6–11] | 8 [6–11] | 0.54 | 3367 |
| ACE-Is/ARBs*† | 192 (47.4) | 1857 (59.5) | <0.001 | 3528 |
| MRAs† | 169 (41.7) | 1420 (45.5) | 0.15 | 3528 |
| Beta-blockers*† | 251 (62.0) | 2112 (67.6) | 0.02 | 3528 |
| Diuretics† | 399 (83.7) | 2616 (83.8) | 0.97 | 3528 |
| Digitalis† | 27 (6.7) | 178 (5.7) | 0.44 | 3528 |
| Pimobendane*† | 36 (8.9) | 153 (4.9) | <0.001 | 3528 |
| Aspirin*† | 128 (31.6) | 1214 (38.9) | 0.005 | 3528 |
| NSAIDs*† | 10 (2.5) | 72 (2.3) | 0.84 | 3528 |
| **Tests at discharge** | | | | |
| BNP, pg/mL | 340 (164–646) | 258 (133–499) | <0.001 | 2241 |
| BNP > 200 pg/mL* | 184 (70.2) | 1211 (61.2) | 0.005 | 2241 |
| NT-proBNP, pg/mL | 3917 [1511–9368] | 1777 [736–3837] | <0.001 | 418 |
| Creatinine, mg/dL | 1.21 (0.86–1.86) | 1.11 (0.86–1.54) | 0.07 | 3000 |
| eGFR, mL/min/1.73m$^2$ | 38.1 (24.2–53.4) | 43.9 (30.4–29.3) | <0.001 | 3484 |
| eGFR <30 mL/min/1.73m$^2$* | 143 (35.9) | 747 (24.2) | <0.001 | 3484 |
| Blood urea nitrogen, mg/dL | 29 (20–42) | 25 (18–35) | <0.001 | 3000 |
| Serum sodium, mEq/L | 139 (136–141) | 139 (137–141) | 0.89 | 3000 |
| Sodium <135 mEq/L | 7 (1.8) | 54 (1.8) | 0.99 | 3000 |
| Hemoglobin, g/dL | 10.8 (9.8–12.1) | 11.4 (9.9–13.0) | <0.001 | 3459 |
| Anemia*†‡ | 315 (78.8) | 2102 (68.7) | <0.001 | 3459 |
| Albumin, g/dL | 3.2 (2.9–3.5) | 3.4 (3.1–3.7) | <0.001 | 3097 |
| Albumin <3.0 g/dL* | 102 (29.3) | 511 (18.6) | <0.001 | 3097 |
| C reactive protein, mg/dL | 0.6 (0.2–1.4) | 0.4 (0.2–1.0) | <0.001 | 3206 |
| C reactive protein > 1.0mg/dL* | 133 (34.8) | 708 (25.1) | <0.001 | 3206 |
| Residual edema* | 150 (37.2) | 297 (9.9) | <0.001 | 3511 |

* Risk adjusting variables selected for the multivariable logistic regression model. Nineteen factors with p-value < 0.1 by univariate analysis were selected.

† Risk adjusting variables selected for the multivariable Cox proportional hazard models.

‡ Defined by the World Health Organization criteria (hemoglobin <12 g/dL for women and <13 g/dL for men).

BMI = body mass index, HF = heart failure, PCI = percutaneous coronary interventions, CABG = coronary artery bypass graft, ACE-I = angiotensin-converting enzyme inhibitor, ARB = angiotensin II receptor blocker, MRA = mineralocorticoid receptor antagonist, HFrEF = heart failure with reduced ejection fraction, HFmrEF = heart failure with mid-range ejection fraction, HFpEF = heart failure with preserved ejection fraction, LVEF = left ventricular ejection fraction, NYHA = New York Heart Association, BNP = brain-type natriuretic peptide, eGFR = estimated glomerular filtration rate, NSAIDs = non-steroidal anti-inflammatory drugs.

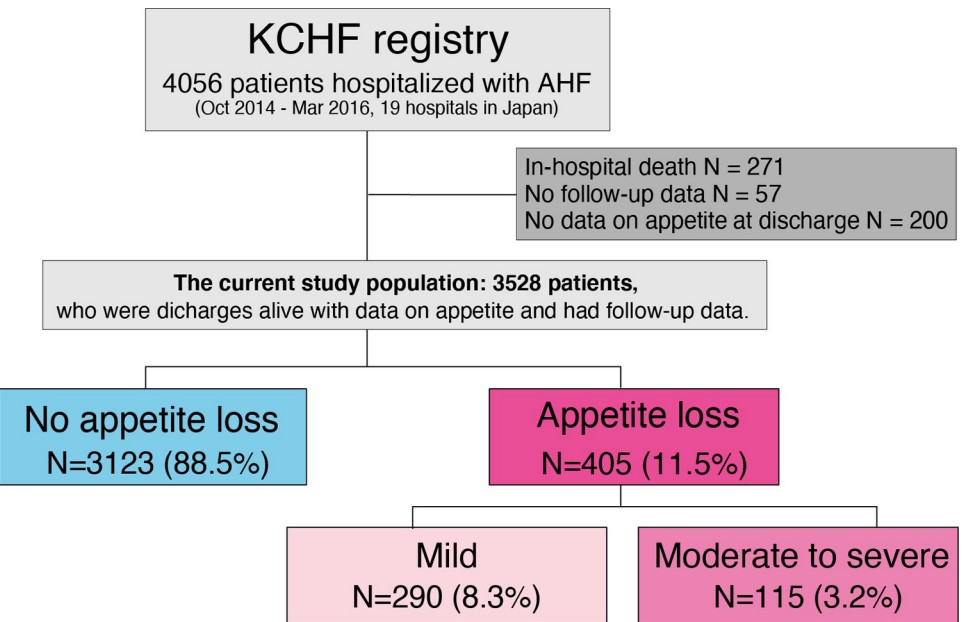

**Fig 1. Study flowchart.** ADHF = acute decompensated heart failure, KCHF = Kyoto Congestive Heart Failure.

(11.5%) with and 3123 patients (88.5%) without appetite loss at hospital discharge (Fig 1). Among 405 patients with appetite loss, 290 patients had mild appetite loss (8.3%) and 115 patients had moderate/severe appetite loss (3.2%) (Fig 1).

## Baseline patient characteristics

Patients with appetite loss were older, more often female, and had a lower BMI than those without appetite loss (Table 1). Chronic kidney disease and dementia were more frequent in patients with appetite loss, whereas hypertension, diabetes, and current smoking were more frequent in those without appetite loss. Ambulatory patients were less frequent in the appetite loss group. LVEF was not significantly different between the two groups. BNP, creatinine, blood urea nitrogen, and CRP levels were significantly higher, whereas sodium and hemoglobin levels were significantly lower in the appetite loss group than in the no appetite loss group. Angiotensin-converting enzyme inhibitor (ACE-I)/angiotensin II receptor blocker (ARB), mineralocorticoid receptor antagonist (MRA), and beta-blockers were less frequently prescribed, whereas pimobendane was prescribed more frequently in the appetite loss group than in the no appetite loss group. The differences in laboratory values at discharge were mostly consistent with those at admission, except for the sodium level. Residual edema was observed more often in the appetite loss group than in the no appetite loss group (Table 1).

## Factors independently associated with appetite loss at discharge

In the multivariable logistic regression analysis, BMI $< 22$ kg/m$^2$ (OR: 1.57, 95% CI: 1.11–2.24, P = 0.01), CRP >1.0mg/dL (OR: 1.49, 95%CI: 1.04–2.14, P = 0.03), and presence of edema at discharge (OR: 4.30, 95%CI: 2.99–6.22, P<0.001) were associated with an increased risk of appetite loss at discharge, whereas ambulatory status (OR: 0.57, 95%CI: 0.39–0.83, P = 0.004) and the use of ACE-I/ARB (OR: 0.70, 95% CI: 0.50–0.98, P = 0.04) were related to a decreased risk in the presence of appetite loss (Table 2). The characteristics of patients who did not use ACE-Is/ARB are shown in S1 Table.

**Table 2. Factors associated with appetite loss at discharge.**

| | Univariate | | | Multivariable | | |
|---|---|---|---|---|---|---|
| | OR | 95%CI | P value | OR | 95%CI | P value |
| Age ≥80 years | 1.72 | 1.39–2.13 | <0.001 | 1.06 | 0.73–1.55 | 0.76 |
| Men | 1.43 | 1.17–1.77 | <0.001 | 1.11 | 0.79–1.56 | 0.53 |
| BMI <22 kg/m$^2$ | 1.65 | 1.33–2.05 | <0.001 | 1.57 | 1.11–2.24 | 0.01 |
| Prior hospitalization due to HF | 1.55 | 1.26–1.92 | <0.001 | 1.14 | 0.82–1.61 | 0.43 |
| Hypertension | 0.70 | 0.56–0.88 | 0.002 | 0.77 | 0.53–1.10 | 0.15 |
| Diabetes mellitus | 0.77 | 0.61–0.96 | 0.018 | 1.34 | 0.92–1.95 | 0.13 |
| Current smoking | 0.53 | 0.36–0.79 | 0.001 | 0.50 | 0.24–1.03 | 0.06 |
| Dementia | 1.87 | 1.47–2.37 | <0.001 | 1.20 | 0.80–1.79 | 0.37 |
| Ambulatory | 0.41 | 0.33–0.51 | <0.001 | 0.57 | 0.39–0.83 | 0.004 |
| BNP >200 pg/mL at discharge | 1.50 | 1.13–1.98 | <0.001 | 1.16 | 0.81–1.67 | 0.41 |
| eGFR <30 mL/min/1.73m$^2$ at discharge | 1.76 | 1.41–2.19 | <0.001 | 1.41 | 0.98–2.04 | 0.06 |
| Anemia at discharge | 1.69 | 1.31–2.17 | 0.006 | 1.27 | 0.85–1.91 | 0.24 |
| C reactive protein >1.0 mg/dL at discharge | 1.60 | 1.27–2.00 | <0.001 | 1.49 | 1.04–2.14 | 0.03 |
| Albumin<3.0mg/dL at discharge | 1.82 | 1.41–2.33 | <0.001 | 1.21 | 0.80–1.83 | 0.36 |
| ACE-Is/ARBs at discharge | 0.61 | 0.50–0.76 | <0.001 | 0.70 | 0.50–0.98 | 0.04 |
| Beta-blockers at discharge | 0.78 | 0.63–0.97 | 0.02 | 0.92 | 0.65–1.30 | 0.63 |
| Aspirin at discharge | 0.73 | 0.58–0.91 | 0.005 | 0.69 | 0.48–0.99 | 0.047 |
| Pimobendane at discharge | 1.89 | 0.30–2.77 | <0.001 | 1.38 | 0.77–2.48 | 0.28 |
| Presence of edema at discharge | 5.61 | 4.44–7.10 | <0.001 | 4.30 | 2.99–6.22 | <0.001 |

BMI = body mass index, ACE-I = angiotensin-converting enzyme inhibitor, ARB = angiotensin II receptor blocker, BNP = brain-type natriuretic peptide, eGFR = estimated glomerular filtration rate, CRP = C reactive protein.

## One-year outcomes: Appetite loss vs. no appetite loss at discharge

The median length of the follow-up was 468 (IQR, 362 to 637) days, with a 95.8% follow-up rate at 1 year. The cumulative 1-year incidence of the primary outcome measure (all-cause death) was significantly higher in patients with appetite loss than in those without appetite loss (31.0% vs. 15.0%, P<0.001) (Fig 2A). The cumulative 1-year incidence of CV death, non-CV death, and HF hospitalization was also significantly higher in patients with appetite loss than in those without appetite loss (19.5% vs. 9.1%, P<0.001; 13.7% vs. 6.0%, P<0.001; and 27.5% vs. 23.5%, P = 0.02, respectively) (Fig 2B–2D). After adjusting for confounders, the excess risk of patients with appetite loss relative to those without appetite loss remained significant for all-cause death, CV death, and non-CV death but was no longer significant for HF hospitalization (Table 3).

In an exploratory analysis, the cumulative 1-year incidence of all-cause death, and CV death was significantly higher in the moderate/severe appetite loss group than in the mild appetite loss group (37.9% vs. 27.8%, P = 0.02; and 28.1% vs. 15.5%, P = 0.002, respectively), whereas the cumulative 1-year incidence of non-CV death and HF hospitalization were not significantly different between the two groups (13.6% vs. 13.7%, P = 0.99; and 26.9% vs. 27.7%, P = 0.88, respectively) (S1A–S1D Fig). In the subgroup analysis, there was a significant interaction between the ambulatory status and the effect of appetite loss on the primary outcome measure (ambulatory: adjusted HR 2.15, 95%CI 1.59–2.90, P<0.001) and non-ambulatory: adjusted HR 1.36, 95%CI 0.94–1.96, P = 0.10, P interaction = 0.02). No interactions were observed for other subgroup factors (S2 Table).

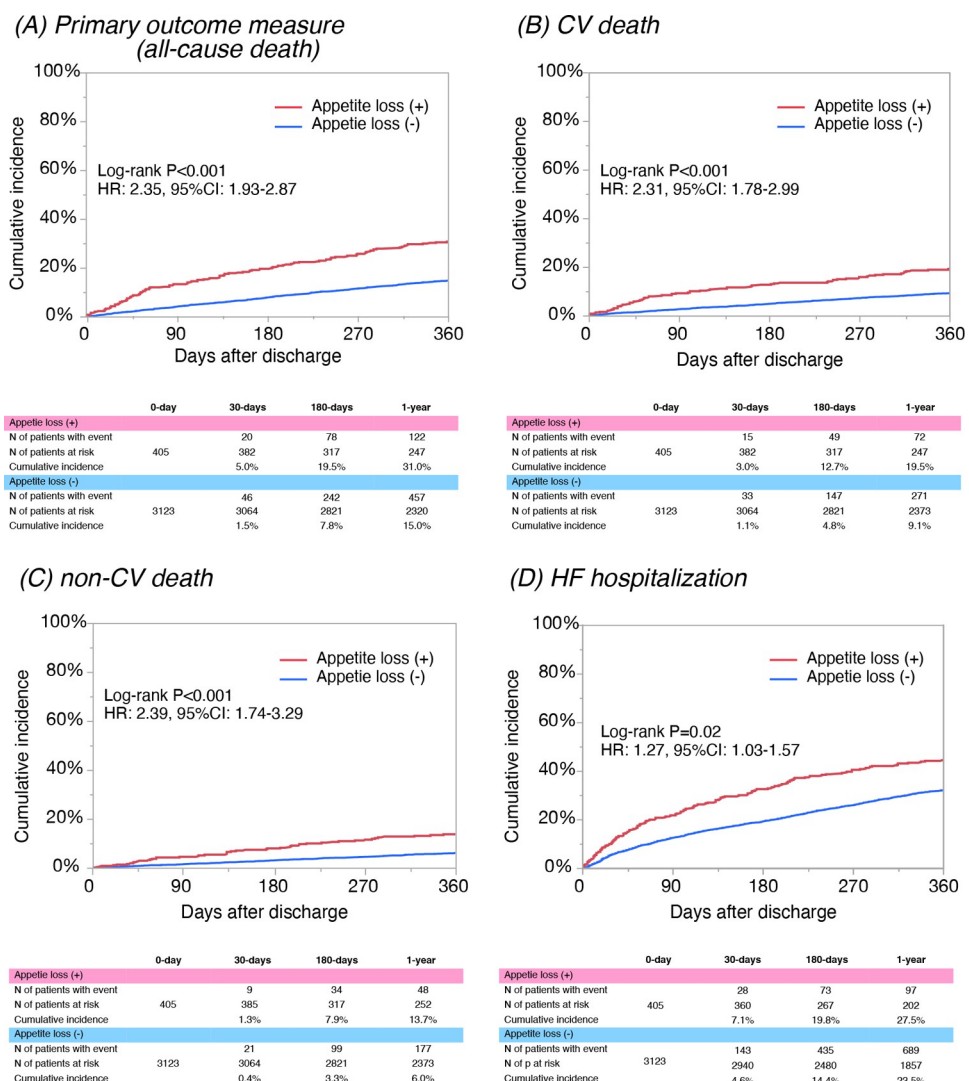

**Fig 2.** Kaplan-Meier curves according to the presence or absence of appetite loss at discharge for (A) the primary outcome measure (all-cause death), (B) CV death, (C) non-CV death, and (D) HF hospitalization. HR, hazard ratio; CI, confidence interval; CV, cardiovascular; HF, heart failure.

## Discussion

In the current study, 11.5% of patients complained appetite loss at discharge. Residual edema, ADL, nutritional status, inflammatory status, and certain types of medication were independently associated with appetite loss. BMI $< 22$ kg/m$^2$, which reflects poor nutritional status, increased the risk of appetite loss incidence, whereas ambulatory status, which is related to higher activity, decreased the risk in the presence of appetite loss. Neither loss nor increase in appetite is common as a side effect of ACE-Is/ARBs, but the positive link between the absence of ACE-Is/ARB use and appetite loss is presumed to be the reflection of renal function and poor general condition of the patients with appetite loss.

Notably, more than 30% of these patients with appetite loss died within 1 year of hospital discharge. This is not surprising because appetite loss is associated not only with well-known prognostic factors in heart failure (such as serum albumin, and hemoglobin), but also with

**Table 3. Clinical outcomes at 1-year.**

| Outcome | Appetite loss (+) | | Appetite loss (-) | | Unadjusted | | | Adjusted | | |
|---|---|---|---|---|---|---|---|---|---|---|
| | N of patients with event /N of patients at risk | Cumulative 1-year incidence | N of patients with event /N of patients at risk | Cumulative 1-year incidence | HR | 95%CI | P-value | HR | 95%CI | P-value |
| **Primary outcome measure** | | | | | | | | | | |
| All-cause death | 122/405 | 31.0% | 457/3123 | 15.0% | 2.35 | 1.93–2.87 | <0.001 | 1.63 | 1.29–2.07 | <0.001 |
| **Secondary outcome measures** | | | | | | | | | | |
| CV death | 73/405 | 19.5% | 274/3123 | 9.2% | 2.31 | 1.78–2.99 | <0.001 | 1.63 | 1.20–2.22 | 0.002 |
| Non-CV death | 48/405 | 13.7% | 177/3123 | 6.0% | 2.39 | 1.74–3.29 | <0.001 | 1.59 | 1.08–2.34 | 0.02 |
| HF hospitalization | 97/405 | 27.5% | 689/3123 | 23.5% | 1.27 | 1.03–1.57 | 0.02 | 0.97 | 0.76–1.23 | 0.78 |

HR = hazard ratio, CI = confidence interval, CV = cardiovascular, HF = heart failure.

factors associated with frailty (cognitive dysfunction and ADL). Decreased serum albumin is not just a sign of nutritional status, but also reflect congestion and advanced pathological changes in the liver due to chronic liver congestion. Anemia in patients with heart failure, as well as hypoalbuminemia, is not only a reflection of nutritional status, but also a result of chronic inflammation caused by heart failure itself, concomitant renal dysfunction, reduced erythropoietin production and sensitivity, and suppression of hematopoietic stem cell function by ACE-Is and ARB.

Previous studies have revealed a close link between the nutritional status and the prognosis of patients with heart failure [1–3,26–29]. The presence of cachexia, frailty, and ADL has also been reported to be associated with prognosis in patients with heart failure [30–32]. In addition, low output due to heart failure leads to intestinal edema and dysfunction. Congestion and hypoalbuminemia due to heart failure also leads to intestinal edema and dysfunction, resulting in loss of appetite. The presence of appetite loss also reflects the severity of heart failure itself, and therefore should be associated with poor long-term outcomes.

In the exploratory analysis, the severity of appetite loss was also associated with all-cause and cardiac deaths. Appetite loss is a subjective symptom, but it is related to outcomes in a severity-dependent manner. The cause-effect relationship between appetite loss and poor clinical outcomes could not be determined because this was an observational study. Therefore, it is unclear whether the improvement in appetite loss is associated with an improvement in nutritional status and clinical outcomes. However, this result implies that appetite is a reliable prognostic marker, and if the patients cannot eat, it is possible that the patients have serious conditions associated with poor prognosis. The current guidelines mention frailty, cachexia, and sarcopenia, as well as iron deficiency [16]. Efficient caloric intake and nutrition supplementation are necessary to solve these problems [33], and it is also important to address loss of appetite. Recently, anamorelin has been used for cancer cachexia [34]. Anamorelin exerts its effects by acting on the ghrelin receptor, GHS-R1a [35]. GHS-R1a is distributed in many tissues and is involved in the release of growth hormone in the pituitary gland and the enhancement of appetite. Anamorelin may have muscle mass and weight gain effects by promoting GH secretion and increasing appetite through the activation of GHS-R1a [35,36]. Although anamorelin is contraindicated in patients with heart failure because of its negative effect on cardiac function, nutritional intervention as well as pharmaceutical intervention may be an option in the future.

## Limitations

The current study has several limitations. First, this was an observational study; therefore, there might be the unmeasured confounders that affect the link between appetite loss and increased mortality. Second, appetite loss is a subjective measure, and there was no objective measure of appetite loss in this study. Third, there was no record of food intake; therefore, it was not possible to compare the actual calorie intake. Fourth, newly initiated medication during hospitalization may affect appetite; however, the appetite date was only available at discharge, and the association with these medications is unknown.

## Conclusions

Loss of appetite at discharge is associated with worse 1-year mortality in patients with ADHF. Appetite is a simple, reliable, and useful subjective marker for risk stratification of patients with ADHF.

## Supporting information

**S1 Fig.** Kaplan-Meier curves according to the degree of appetite loss at discharge for (A) the primary outcome measure (all-cause death), (B) CV death, (C) non-CV death, and (D) HF hospitalization. HR = hazard ratio, CI = confidence interval, CV = cardiovascular, HF = heart failure.
(PDF)

**S1 Table. Patient characteristics without ACE-Is/ARB use.**
(PDF)

**S2 Table. Subgroup analysis for the effect of appetite loss on the primary outcome measure.**
(PDF)

**S1 File. This contains the supplementary methods.**
(PDF)

## Acknowledgments

The members of the KCHF study investigators are as follows: Erika Yamamoto, MD, Hidenori Yaku, MD, Takao Kato, MD, Neiko Ozasa, MD, Yusuke Yoshikawa, MD, Tetsuo Shioi, MD, Koichiro Kuwahara, MD, and Takeshi Kimura, MD, Kyoto University Hospital, Kyoto, Japan; Moritake Iguchi, MD, Yugo Yamashita, MD, and Masaharu Akao, MD, National Hospital Organization Kyoto Medical Center, Kyoto, Japan; Masashi Kato, MD, and Shinji Miki, MD, Mitsubishi Kyoto Hospital, Kyoto, Japan; Mamoru Takahashi, MD, Shimabara Hospital, Kyoto, Japan; Tsuneaki Kawashima, MD, and Takafumi Yagi, MD, Daini Okamoto General Hospital, Kyoto, Japan; Toshikazu Jinnai, MD, and Takashi Konishi, MD, Japanese Red Cross Otsu Hospital, Otsu, Japan; Yasutaka Inuzuka, MD, and Shigeru Ikeguchi, MD, Shiga General Hospital, Moriyama, Japan; Tomoyuki Ikeda, MD, and Yoshihiro Himura, MD, Hikone Municipal Hospital, Hikone, Japan; Kazuya Nagao, MD, and Tsukasa Inada, MD, Osaka Red Cross Hospital, Osaka, Japan; Kenichi Sasaki, MD, and Moriaki Inoko, MD, Kitano Hospital, Osaka, Japan; Takafumi Kawai, MD, Tomoki Sasa, MD, and Mitsuo Matsuda, MD, Kishiwada City Hospital, Kishiwada, Japan; Akihiro Komasa, MD, and Katsuhisa Ishii, MD, Kansai Electric Power Hospital, Osaka, Japan; Yodo Tamaki, MD, and Yoshihisa Nakagawa, MD, Tenri Hospital, Tenri, Japan; Ryoji Taniguchi, MD, Yukihito Sato, MD, and Yoshiki Takatsu, MD,

Hyogo Prefectural Amagasaki General Medical Center, Amagasaki, Japan; Takeshi Kitai, MD, Ryousuke Murai, MD, and Yutaka Furukawa, MD, Kobe City Medical Center General Hospital, Kobe, Japan; Mitsunori Kawato, MD, Kobe City NishiKobe Medical Center, Kobe, Japan; Yasuyo Motohashi, MD, Kanae Su, MD, Mamoru Toyofuku, MD, and Takashi Tamura, MD, Japanese Red Cross Wakayama Medical Center, Wakayama, Japan; Reiko Hozo, MD, Ryusuke Nishikawa, MD, and Hiroki Sakamoto, MD, Shizuoka General Hospital, Shizuoka, Japan; Yuichi Kawase, MD, Keiichiro Iwasaki, MD, and Kazushige Kadota, MD, Kurashiki Central Hospital, Kurashiki, Japan; and Takashi Morinaga, MD, Yohei Kobayashi, MD, and Kenji Ando, MD, Kokura Memorial Hospital, Kokura, Japan.

The lead author is Erika Yamamoto, MD <erkymmt@kuhp.kyoto-u.ac.jp>.

## Author Contributions

**Conceptualization:** Erika Yamamoto, Takao Kato, Hidenori Yaku, Yasutaka Inuzuka, Yodo Tamaki, Neiko Ozasa, Takashi Morinaga.

**Data curation:** Erika Yamamoto, Takao Kato, Hidenori Yaku, Yasutaka Inuzuka, Yodo Tamaki, Neiko Ozasa, Yusuke Yoshikawa, Takeshi Kitai, Ryoji Taniguchi, Moritake Iguchi, Masashi Kato, Mamoru Takahashi, Toshikazu Jinnai, Tomoyuki Ikeda, Kazuya Nagao, Takafumi Kawai, Akihiro Komasa, Ryusuke Nishikawa, Yuichi Kawase, Takashi Morinaga, Mitsunori Kawato, Yuta Seko, Masayuki Shiba, Mamoru Toyofuku, Yutaka Furukawa, Yoshihisa Nakagawa, Kenji Ando, Kazushige Kadota, Satoshi Shizuta, Koh Ono, Yukihito Sato, Koichiro Kuwahara.

**Formal analysis:** Erika Yamamoto, Takeshi Morimoto.

**Investigation:** Erika Yamamoto, Takao Kato, Hidenori Yaku, Yasutaka Inuzuka, Yodo Tamaki, Neiko Ozasa, Yusuke Yoshikawa, Takeshi Kitai, Ryoji Taniguchi, Moritake Iguchi, Masashi Kato, Mamoru Takahashi, Toshikazu Jinnai, Tomoyuki Ikeda, Kazuya Nagao, Takafumi Kawai, Akihiro Komasa, Ryusuke Nishikawa, Yuichi Kawase, Takashi Morinaga, Mitsunori Kawato, Yuta Seko, Masayuki Shiba, Mamoru Toyofuku, Yutaka Furukawa, Yoshihisa Nakagawa, Kenji Ando, Kazushige Kadota, Satoshi Shizuta, Koh Ono, Yukihito Sato, Koichiro Kuwahara.

**Methodology:** Erika Yamamoto, Takao Kato, Hidenori Yaku, Takeshi Morimoto, Yasutaka Inuzuka, Yodo Tamaki, Neiko Ozasa, Yusuke Yoshikawa, Takeshi Kitai, Ryoji Taniguchi, Moritake Iguchi, Masashi Kato, Mamoru Takahashi, Toshikazu Jinnai, Tomoyuki Ikeda, Kazuya Nagao, Takafumi Kawai, Akihiro Komasa, Ryusuke Nishikawa, Yuichi Kawase, Takashi Morinaga, Mitsunori Kawato, Yuta Seko, Masayuki Shiba, Mamoru Toyofuku, Yutaka Furukawa, Yoshihisa Nakagawa, Kenji Ando, Kazushige Kadota, Satoshi Shizuta, Koh Ono, Yukihito Sato, Koichiro Kuwahara, Takeshi Kimura.

**Project administration:** Erika Yamamoto, Takao Kato, Hidenori Yaku, Takeshi Morimoto, Yasutaka Inuzuka, Yodo Tamaki, Neiko Ozasa, Takeshi Kimura.

**Supervision:** Takeshi Kimura.

**Writing – original draft:** Erika Yamamoto.

**Writing – review & editing:** Takao Kato, Hidenori Yaku, Takeshi Morimoto, Yasutaka Inuzuka, Yodo Tamaki, Neiko Ozasa, Yusuke Yoshikawa, Takeshi Kitai, Ryoji Taniguchi, Moritake Iguchi, Masashi Kato, Mamoru Takahashi, Toshikazu Jinnai, Tomoyuki Ikeda, Kazuya Nagao, Takafumi Kawai, Akihiro Komasa, Ryusuke Nishikawa, Yuichi Kawase, Takashi Morinaga, Mitsunori Kawato, Yuta Seko, Masayuki Shiba, Mamoru Toyofuku,

Yutaka Furukawa, Yoshihisa Nakagawa, Kenji Ando, Kazushige Kadota, Satoshi Shizuta, Koh Ono, Yukihito Sato, Koichiro Kuwahara, Takeshi Kimura.

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
