## [Decision Letter · Decision Letter 0]

15 Nov 2021

PONE-D-21-32941Appetite Loss at Discharge from Acute Decompensated Heart Failure: Observation from KCHF registryPLOS ONE

Dear Dr. Kato,

Thank you for submitting your manuscript to PLOS ONE. After careful consideration, we feel that it has merit but does not fully meet PLOS ONE’s publication criteria as it currently stands. Therefore, we invite you to submit a revised version of the manuscript that addresses the points raised during the review process.

ACADEMIC EDITOR: Serious methodological issues weaken the manuscript and they shoul be addressed to better support the conclusions. All issues highlighted by revierwers are required.

We look forward to receiving your revised manuscript.

Kind regards,

Vincenzo Lionetti, M.D., PhD

Academic Editor

PLOS ONE

Journal Requirements:

2. One of the noted authors is a group or consortium KCHF study investigators. In addition to naming the author group, please list the individual authors and affiliations within this group in the acknowledgments section of your manuscript. Please also indicate clearly a lead author for this group along with a contact email address.

Reviewers' comments:

Reviewer's Responses to Questions

**Comments to the Author**

1. Is the manuscript technically sound, and do the data support the conclusions?

Reviewer #1: Yes

Reviewer #2: Yes

2. Has the statistical analysis been performed appropriately and rigorously? 

Reviewer #1: Yes

Reviewer #2: Yes

3. Have the authors made all data underlying the findings in their manuscript fully available?

Reviewer #1: Yes

Reviewer #2: Yes

4. Is the manuscript presented in an intelligible fashion and written in standard English?

Reviewer #1: Yes

Reviewer #2: Yes

5. Review Comments to the Author

Reviewer #1: A paper of Yamamoto et al. presents results of an observational study which included over 3 thousand patients with acute heart failure. The study population and follow-up time (mean 1-2 years) are the great advantages of the study. Methods and results are clearly described. Unfortunately, discussion is very simplistic with only a superficial analysis of potential mechanisms linking appetite loss and heart failure. Moreover, Authors haven’t referred any original papers concerning the issue of appetite loss and anorexia/cachexia syndrome in heart failure (and in the past years there were at least few important papers published). First paragraph of the discussion is in the fact repeated results’ summary which, in my opinion, is needless.

Furthermore I have some minor objections:

1. Phrase “ambulatory status” in the abstract is unclear – did the Authors meant “ambulatory” vs “hospitalized” or NYHA class? Then in the result section again is an information that “Ambulatory patients were less frequent in the appetite loss group” – it should a status at hospital discharge so what is the “non-ambulatory status”? I think it should be defined

2. Where are the results of subgroup analysis depending on “activity of daily living”?

3. “Neither loss nor increase of appetite is common as a side effect of ACE-Is/ARBs, but the

positive link between the absence of ACE-Is/ARB use and appetite loss is presumed to be the

reflection of renal function and poor general conditions of the patients with appetite loss.” – there is not direct data presented in the study which could support the thesis

4. Decreased serum albumin and hemoglobin concentration are not just a sign of nutrition status but reflect advanced pathological changes in heart failure, including i.a. liver and renal failure, iron deficiency

Reviewer #2: This is a retrospective analysis from the Kyoto Congestive Heart Failure registry focusing on the prevalence and prognostic impact of appetite loss in 3,528 patients with acute heart failure. As one might expect, appetite loss at discharge was associated with worse clinical status and a worse 1-year outcome. The study is original in that it explores appetite loss in this setting. On the other hand, it has the limitations of a retrospective analysis on a non-dedicated database, namely the lack of relevant information such as food intake etc. Study limitations are correctly acknowledged by the Authors. Please find below some other comments.

Patients were not selected based on their cognitive status, and assessment of appetite loss (as well as its classification as mild, moderate or severe) relied exclusively on patient report. Food intake and calories per day could have been easily checked in hospitalized patients, and would have provided objective metrics for analysis.

Please specify more clearly what you mean by "ambulatory status" (patients treated with iv diuretics without hospitalization?)

Apart from medications at discharge, you should have considered also the start of drugs from admission to discharge, as starting multiple therapies could reduce patient appetite.

Table 2 reports only univariable predictors. Please provide the full list of variables evaluated as possible univariable predictors. It is important to understand if relevant variables such as the length of hospitalization were taken into account. The list of univariable predictors is rather short, with several arbitrarily dichotomized variables.

In the abstract, please specify the variables included in the multivariable model.

The approval numbers by the many centers (page 12) could be moved to the supplementary material.

The manuscript would benefit from linguistic revision.

6. PLOS authors have the option to publish the peer review history of their article (what does this mean?). If published, this will include your full peer review and any attached files.

Reviewer #1: **Yes: **Anna Drohomirecka

Reviewer #2: No

---

## [Author Response · Author response to Decision Letter 0]

25 Dec 2021

Reviewer #1: A paper of Yamamoto et al. presents results of an observational study which included over 3 thousand patients with acute heart failure. The study population and follow-up time (mean 1-2 years) are the great advantages of the study. Methods and results are clearly described. Unfortunately, discussion is very simplistic with only a superficial analysis of potential mechanisms linking appetite loss and heart failure. 

Thank you so much for the important comment. Actually, the analysis is very simple in this study. KCHF study is not just focused on the nutrition and heart failure, therefore we didn’t collect many important factors, such as the change of body weight, or any objective evaluation measure about appetite loss (CNAQ-J, MNA etc.) and cannot draw any conclusion about the causal relationship between appetite, nutrition status, and prognosis. We would like to avoid overstating the result and just show that the patient with appetite loss at hospital discharge was associated with poor outcome. Based on this situation, we added several sentences in discussion part. 

The current guidelines mention frailty, cachexia, and sarcopenia, as well as iron deficiency.12 Efficient caloric intake and nutrition supplementation is necessary to solve these problems, and it is also important to address loss of appetite. Recently, Anamorelin has been used for cancer cachexia. Anamorelin exerts its effects by acting on the ghrelin receptor, GHS-R1a. GHS-R1a is distributed in many tissues and is involved in the release of growth hormone in the pituitary gland and the enhancement of appetite. Anamorelin may have muscle mass and weight gain effects by promoting GH secretion and increasing appetite through activation of GHS-R1a. Although Anamorelin is contraindicated in heart failure patients because of its negative effect for cardiac function, pharmaceutical intervention as well as nutritional intervention may be an option in the future.

Moreover, Authors haven’t referred any original papers concerning the issue of appetite loss and anorexia/cachexia syndrome in heart failure (and in the past years there were at least few important papers published). 

Thank you so much for your constructive comment. We added several references. 

First paragraph of the discussion is in the fact repeated results’ summary which, in my opinion, is needless.

Thank you so much for the comment. We deleted summary in discussion.

1. Phrase “ambulatory status” in the abstract is unclear – did the Authors meant “ambulatory” vs “hospitalized” or NYHA class? Then in the result section again is an information that “Ambulatory patients were less frequent in the appetite loss group” – it should a status at hospital discharge so what is the “non-ambulatory status”? I think it should be defined.

Thank you so much for the comment. We defined ambulatory status as patients who can walk by themselves. In this study, we assessed patient’s ADL on four level: (1) ambulatory (patients who can walk by themselves), (2) use wheelchair (outdoor only), (3) use wheelchair (outdoor and indoor), (4) bedridden. “Ambulatory status” means patients with highest ADL level among them. We added definition in supplementary appendix.

2. Where are the results of subgroup analysis depending on “activity of daily living”?

Thank you so much for the comment. As mentioned above, “ambulatory” means ADL status. “ambulatory” vs “non-ambulatory” (the third in supplementary table 1) correspond the result of subgroup analysis depending on ADL.

3. “Neither loss nor increase of appetite is common as a side effect of ACE-Is/ARBs, but the positive link between the absence of ACE-Is/ARB use and appetite loss is presumed to be the reflection of renal function and poor general conditions of the patients with appetite loss.” – there is not direct data presented in the study which could support the thesis.

Thank you so much for the constructive comment. We evaluated whether general condition, including renal function is worse in patients without ACE-Is/ARB use. We have added the Characteristics of patients without ACE-Is/ARB use in the Supplementary Table 1.

　 　 Appetite loss (-) 　 　 　 Appetite loss (+) 　

　 ACE-I/ARB (+) ACE-I/ARB (-) P 　 ACE-I/ARB (+) ACE-I/ARB (-) P

Age 78 (69-85) 82 (74-87) <0.001 　 82 (73-87) 84 (78-89) 0.02

Age ≧80 843 (45.4) 725 (57.3) <0.001 　 111 (57.8) 146 (68.5) 0.03

Male 1098 (59.2) 663 (52.4) <0.001 　 97 (50.5) 95 (44.6) 0.23

BMI≦22 750 (41.8) 587 (49.2) <0.001 　 98 (54.8) 116 (59.5) 0.35

hx of HF 623 (33.9) 451 (36.3) 0.17 　 69 (38.1) 108 (51.7) 0.007

Anemia 1123 (60.5) 910 (72.1) <0.001 　 126 (66.6) 166 (77.9) 0.006

Alb<3.0mg/dL 189 (10.4) 188 (15.4) <0.001 　 34 (18.5) 40 (19.6) 0.78

eGFR<30 355 (19.1) 421 (33.3) <0.001 　 55 (28.7) 83 (39.2) 0.03

Na<135 177 (9.6) 171 (13.6) <0.001 　 21 (10.9) 39 (18.5) 0.03

CRP>1.0 369 (22.2) 339 (29.1) <0.001 　 79 (64.8) 105 (75.0) 0.07

HFrEF 769 (41.5) 383 (30.4) <0.001 　 79 (41.4) 63 (29.6) 0.01

Patients without ACE-I/ARB were older, had lower BMI and worse renal function compared to patients with ACE-I/ARB in both patients with and without appetite loss.

4. Decreased serum albumin and hemoglobin concentration are not just a sign of nutrition status but reflect advanced pathological changes in heart failure, including i.a. liver and renal failure, iron deficiency

Thank you so much for the comment. We agree with author’s comment. We added the sentence below in discussion part.

“Decreased serum albumin is not just a sign of nutrition status but reflect congestion and advanced pathological changes in liver due to chronic liver congestion. Anemia in patients with heart failure, as well as hypoalbuminemia, is not only a reflection of nutritional status, but also a result of chronic inflammation caused by heart failure itself, concomitant renal dysfunction and reduced erythropoietin production and sensitivity, as well as suppression of hematopoietic stem cell function by ACE inhibitors and angiotensin II receptor blockers.”

Reviewer #2: This is a retrospective analysis from the Kyoto Congestive Heart Failure registry focusing on the prevalence and prognostic impact of appetite loss in 3,528 patients with acute heart failure. As one might expect, appetite loss at discharge was associated with worse clinical status and a worse 1-year outcome. The study is original in that it explores appetite loss in this setting. On the other hand, it has the limitations of a retrospective analysis on a non-dedicated database, namely the lack of relevant information such as food intake etc. Study limitations are correctly acknowledged by the Authors. Please find below some other comments.

Patients were not selected based on their cognitive status, and assessment of appetite loss (as well as its classification as mild, moderate or severe) relied exclusively on patient report. Food intake and calories per day could have been easily checked in hospitalized patients, and would have provided objective metrics for analysis.

Thank you for the constructive comment. We would like to collect these quantitative data in next study. 

Please specify more clearly what you mean by "ambulatory status" (patients treated with iv diuretics without hospitalization?)

Thank you for the comment. We defined ambulatory status as patients who can walk by themselves. In this study, we assessed patient’s ADL on four level: (1) ambulatory (patients who can walk by themselves), (2) use wheelchair (outdoor only), (3) use wheelchair (outdoor and indoor), (4) bedridden. “Ambulatory status” means patients with highest ADL level among them. We added definition in supplementary appendix.

Apart from medications at discharge, you should have considered also the start of drugs from admission to discharge, as starting multiple therapies could reduce patient appetite.

Thank you for pointing out important point. We added the sentence below in limitation part.

“Forth, newly initiated medication during hospitalization may affect appetite, however appetite date was only available at discharge, and the association with those medication is unknown.” 

Table 2 reports only univariable predictors. Please provide the full list of variables evaluated as possible univariable predictors. It is important to understand if relevant variables such as the length of hospitalization were taken into account. The list of univariable predictors is rather short, with several arbitrarily dichotomized variables.

Thank you for the comment. We assessed clinical and laboratory data shown in Table 1, and choose 19 factors with P value < 0.1 by univariate analysis. We added the underlined part in method section.

A multivariable logistic regression model was developed to identify clinical characteristics, and laboratory data and medications at discharge that were independently associated with the presence of appetite loss using 19 factors with P value < 0.1 by univariate analysis extracted from the variables listed in Table 1.” 

In the abstract, please specify the variables included in the multivariable model.

Thank you for the comment. We added the underlined part in abstract.

“In the multivariable logistic regression analysis using 19 clinical and laboratory factors with P value < 0.1 by univariate analysis, BMI…”

The approval numbers by the many centers (page 12) could be moved　to the supplementary material.

Thank you for the comment. We moved the individual approval numbers to supplementary appendix. 

The manuscript would benefit from linguistic revision.

Thank you for the constructive comment. The revised manuscript has been edited by native speaker.

---

## [Decision Letter · Decision Letter 1]

13 Jan 2022

PONE-D-21-32941R1Appetite Loss at Discharge from Acute Decompensated Heart Failure: Observation from KCHF registryPLOS ONE

Dear Dr. Kato,

Thank you for submitting your manuscript to PLOS ONE. After careful consideration, we feel that it has merit but does not fully meet PLOS ONE’s publication criteria as it currently stands. Therefore, we invite you to submit a revised version of the manuscript that addresses the points raised during the review process.

ACADEMIC EDITOR: All issues raised by expert reviewers are required.

We look forward to receiving your revised manuscript.

Kind regards,

Vincenzo Lionetti, M.D., PhD

Academic Editor

PLOS ONE

Journal Requirements:

Reviewers' comments:

Reviewer's Responses to Questions

**Comments to the Author**

1. If the authors have adequately addressed your comments raised in a previous round of review and you feel that this manuscript is now acceptable for publication, you may indicate that here to bypass the “Comments to the Author” section, enter your conflict of interest statement in the “Confidential to Editor” section, and submit your "Accept" recommendation.

Reviewer #1: (No Response)

2. Is the manuscript technically sound, and do the data support the conclusions?

Reviewer #1: Yes

3. Has the statistical analysis been performed appropriately and rigorously? 

Reviewer #1: Yes

4. Have the authors made all data underlying the findings in their manuscript fully available?

Reviewer #1: Yes

5. Is the manuscript presented in an intelligible fashion and written in standard English?

Reviewer #1: Yes

6. Review Comments to the Author

Reviewer #1: The revised manuscript replies to most but not all comments of the previous review. Authors significantly improved the discussion section and clarified some ambiguities. Nevertheless, in my opinion, some parts still needs explanation or completion.

The definition of “ambulatory status” should be placed in the main text body not in the supplement.

A statement “However, there was no study focused on “appetite loss,” which is a

subjective symptom in patients with heart failure. “ is untrue – there are studies concerned on appetite loss in patient with heart failure and Authors have been already asked to quote these in the discussion.

There are sentences which need rephrasing:

- “We assessed 3528 patients who were alive at discharge for whom appetite and had follow-up data were available.”

- “In the multivariable logistic regression analysis using 19 clinical and laboratory factors with P value < 0.1 by univariate analysis, BMI < 22 kg/m2 (odds ratio (OR): 1.57, 95% confidence interval (CI): 1.11-2.24, P=0.01), CRP >1.0mg/dL (OR: 1.49, 95%CI: 1.04-2.14, P=0.03), and presence of edema at discharge (OR: 4.30, 95%CI: 2.99-6.22, P<0.001) were positively associated with appetite loss at discharge, whereas ambulatory status (OR: 0.57, 95%CI: 0.39-0.83, P=0.004) and the use of ACE-I/ARB (OR: 0.70, 95% CI: 0.50-0.99, P=0.04) were negatively associated with appetite loss.” Phrase “positive/negative association” when odds ratio are used is a little bit unfortunate because it is not a correlation but risk increase/decrease.

- “The details have been previously descrived.” – please correct a typing mistake

- “When we consider the cause of appetite loss, this result is clinically important, because there is no argument about the association between the amount of activity, nutritional status, and the presence of appetite loss.” – the sentence is either unclear or false. There are many links between amount of activity, nutritional status and appetite loss.

- “In addition, low output due to heart failure leads intestinal edema and dysfunction.

Congestion and hypoalbuminemia due to heart failure also leads intestinal edema and

dysfunction, resulting in loss of appetite.” or “leads to” ?

7. PLOS authors have the option to publish the peer review history of their article (what does this mean?). If published, this will include your full peer review and any attached files.

Reviewer #1: **Yes: **Anna Drohomirecka

---

## [Author Response · Author response to Decision Letter 1]

31 Jan 2022

Reviewer #1: The revised manuscript replies to most but not all comments of the previous review. Authors significantly improved the discussion section and clarified some ambiguities. Nevertheless, in my opinion, some parts still needs explanation or completion.

We appreciate the time and effort that you have dedicated to providing your valuable feedback on my manuscript. We revised the manuscript again according to your comments.

The definition of “ambulatory status” should be placed in the main text body not in the supplement.

Thank you so much for the comment. We moved the definition of “ambulatory status” in the supplement to main text.

A statement “However, there was no study focused on “appetite loss,” which is a subjective symptom in patients with heart failure. “ is untrue – there are studies concerned on appetite loss in patient with heart failure and Authors have been already asked to quote these in the discussion.

Thank you so much for the comment. Based on your comment, we have revised the sentence as follows. 

P3 L15 Although in the past years there were at least few important papers published concerning the issue of appetite loss and anorexia/cachexia syndrome in heart failure [PMID: 9107242, PMID: 25704364, PMID: 34757487, PMID: 24347122, PMID: 26865478], there is still scarcity in the studies focused on “appetite loss,” a subjective symptom in patients with heart failure [PMID: 24347122, PMID: 26865478].

There are sentences which need rephrasing:

- “We assessed 3528 patients who were alive at discharge for whom appetite and had follow-up data were available.”

Thank you so much for the comment. Based on your comment, we have revised the sentence as follows.

P2 L7 We assessed 3528 patients alive at discharge, and for whom appetite and follow-up data were available.

- “In the multivariable logistic regression analysis using 19 clinical and laboratory factors with P value < 0.1 by univariate analysis, BMI < 22 kg/m2 (odds ratio (OR): 1.57, 95% confidence interval (CI): 1.11-2.24, P=0.01), CRP >1.0mg/dL (OR: 1.49, 95%CI: 1.04-2.14, P=0.03), and presence of edema at discharge (OR: 4.30, 95%CI: 2.99-6.22, P<0.001) were positively associated with appetite loss at discharge, whereas ambulatory status (OR: 0.57, 95%CI: 0.39-0.83, P=0.004) and the use of ACE-I/ARB (OR: 0.70, 95% CI: 0.50-0.99, P=0.04) were negatively associated with appetite loss.” Phrase “positive/negative association” when odds ratio are used is a little bit unfortunate because it is not a correlation but risk increase/decrease.

Thank you so much for the comment. Based on your comment, we have revised the sentence as follows.

P2 L9 In the multivariable logistic regression analysis using 19 clinical and laboratory factors with P value < 0.1 by univariate analysis, BMI < 22 kg/m2 (odds ratio (OR): 1.57, 95% confidence interval (CI): 1.11-2.24, P=0.01), CRP >1.0mg/dL (OR: 1.49, 95%CI: 1.04-2.14, P=0.03), and presence of edema at discharge (OR: 4.30, 95%CI: 2.99-6.22, P<0.001) were associated with an increased risk of appetite loss at discharge, whereas ambulatory status (OR: 0.57, 95%CI: 0.39-0.83, P=0.004) and the use of ACE-I/ARB (OR: 0.70, 95% CI: 0.50-0.99, P=0.04) were related to decrease risk in the presence of appetite loss.

- “The details have been previously descrived.” – please correct a typing mistake

Thank you so much for the comment. Based on your comment, we have revised the sentence as follows.

P5 L9 The details have been previously described.

- “When we consider the cause of appetite loss, this result is clinically important, because there is no argument about the association between the amount of activity, nutritional status, and the presence of appetite loss.” – the sentence is either unclear or false. There are many links between amount of activity, nutritional status and appetite loss.

Thank you so much for the comment. As you pointed out, we also think there are many links between these factors and appetite loss and the result of this study are reasonable. We revised the sentence more simply.

P19 L14 When we consider the cause of appetite loss, there is no argument about the association between the amount of activity, nutritional status, and the presence of appetite loss.

- “In addition, low output due to heart failure leads intestinal edema and dysfunction.

Congestion and hypoalbuminemia due to heart failure also leads intestinal edema and

dysfunction, resulting in loss of appetite.” or “leads to” ?

Thank you so much for the comment. Based on your comment, we have revised the sentence as follows.

P20L5 In addition, low output due to heart failure leads to intestinal edema and dysfunction. Congestion and hypoalbuminemia due to heart failure also leads to intestinal edema and dysfunction, resulting in loss of appetite.

---

## [Decision Letter · Decision Letter 2]

18 Mar 2022

PONE-D-21-32941R2Appetite Loss at Discharge from Acute Decompensated Heart Failure: Observation from KCHF registryPLOS ONE

Dear Dr. Kato,

Thank you for submitting your manuscript to PLOS ONE. After careful consideration, we feel that it has merit but does not fully meet PLOS ONE’s publication criteria as it currently stands. Therefore, we invite you to submit a revised version of the manuscript that addresses the points raised during the review process.

ACADEMIC EDITOR: All issues raised by expert reviewer are required.

We look forward to receiving your revised manuscript.

Kind regards,

Vincenzo Lionetti, M.D., PhD

Academic Editor

PLOS ONE

Journal Requirements:

Reviewers' comments:

Reviewer's Responses to Questions

**Comments to the Author**

1. If the authors have adequately addressed your comments raised in a previous round of review and you feel that this manuscript is now acceptable for publication, you may indicate that here to bypass the “Comments to the Author” section, enter your conflict of interest statement in the “Confidential to Editor” section, and submit your "Accept" recommendation.

Reviewer #1: (No Response)

2. Is the manuscript technically sound, and do the data support the conclusions?

Reviewer #1: Yes

3. Has the statistical analysis been performed appropriately and rigorously? 

Reviewer #1: Yes

4. Have the authors made all data underlying the findings in their manuscript fully available?

Reviewer #1: Yes

5. Is the manuscript presented in an intelligible fashion and written in standard English?

Reviewer #1: Yes

6. Review Comments to the Author

Reviewer #1: The majority of concerns raised in the previous review were addressed. However, there are a few questions left.

I still have doubts about the following conclusion:

“When we consider the cause of appetite loss, because there is no argument about the association between the amount of activity, nutritional status, and the presence of appetite loss.”

It stands in the opposition to the results: BMI < 22 kg/m2 increased the risk of appetite loss incidence (OR: 1.57, 95% CI: 1.11-2.24, P=0.01), whereas ambulatory status (OR: 0.57, 95% CI: 0.39-0.83, P=0.004) decreased. BMI reflect nutritional status, ambulatory status is related to higher activity.

Authors still used the phrase “positively/negatively associated” in the context of risk assessment:

“In the multivariable logistic regression analysis, BMI < 22 kg/m2 (OR: 1.57, 95% CI: 1.11-

2.24, P=0.01), CRP > 1.0mg/dL (OR: 1.49, 95% CI: 1.04-2.14, P=0.03), and the presence of edema at discharge (OR: 4.30, 95% CI: 12.99-6.22, P<0.001) were positively associated with appetite loss at discharge, whereas ambulatory status (OR: 0.57, 95% CI: 0.39-0.83, P=0.004) and the use of ACE-I/ARB (OR:0.70, 95% CI: 0.50-0.99, P=0.04) were negatively associated with appetite loss (Table 2).” (page 14)

There is a typing mistake in the abstract:

“were related to decrease risk in the presence of appetite loss” instead of: “were related to decreased risk in the presence of appetite loss

7. PLOS authors have the option to publish the peer review history of their article (what does this mean?). If published, this will include your full peer review and any attached files.

Reviewer #1: **Yes: **Anna Drohomirecka

---

## [Author Response · Author response to Decision Letter 2]

18 Mar 2022

Response:

We thank the reviewer for detailed assessment and positive comments. We have revised the manuscript according to your suggestions.

Reviewer #1: The majority of concerns raised in the previous review were addressed. However, there are a few questions left.

I still have doubts about the following conclusion:

“When we consider the cause of appetite loss, because there is no argument about the association between the amount of activity, nutritional status, and the presence of appetite loss.”

It stands in the opposition to the results: BMI < 22 kg/m2 increased the risk of appetite loss incidence (OR: 1.57, 95% CI: 1.11-2.24, P=0.01), whereas ambulatory status (OR: 0.57, 95% CI: 0.39-0.83, P=0.004) decreased. BMI reflect nutritional status, ambulatory status is related to higher activity.

Response:

Thank you for the important comments. As you pointed out, poor nutritional status increased the risk of appetite loss incidence, whereas higher activity decreased the risk of appetite loss incidence. We have revised the manuscript as follows: 

“BMI < 22 kg/m2, which reflects poor nutritional status, increased the risk of appetite loss incidence, whereas ambulatory status, which is related to higher activity, decreased the risk in the presence of appetite loss.” (Page 20, Lines 4-6)

Authors still used the phrase “positively/negatively associated” in the context of risk assessment:

“In the multivariable logistic regression analysis, BMI < 22 kg/m2 (OR: 1.57, 95% CI: 1.11-

2.24, P=0.01), CRP > 1.0mg/dL (OR: 1.49, 95% CI: 1.04-2.14, P=0.03), and the presence of edema at discharge (OR: 4.30, 95% CI: 12.99-6.22, P<0.001) were positively associated with appetite loss at discharge, whereas ambulatory status (OR: 0.57, 95% CI: 0.39-0.83, P=0.004) and the use of ACE-I/ARB (OR:0.70, 95% CI: 0.50-0.99, P=0.04) were negatively associated with appetite loss (Table 2).” (page 14)

Response:

We appreciate your comments. We have deleted the phrases “positively/negatively associated”. We have changed these as follows:

“In the multivariable logistic regression analysis, BMI < 22 kg/m2 (OR: 1.57, 95% CI: 1.11-2.24, P=0.01), CRP >1.0mg/dL (OR: 1.49, 95%CI: 1.04-2.14, P=0.03), and presence of edema at discharge (OR: 4.30, 95%CI: 2.99-6.22, P<0.001) were associated with an increased risk of appetite loss at discharge, whereas ambulatory status (OR: 0.57, 95%CI: 0.39-0.83, P=0.004) and the use of ACE-I/ARB (OR: 0.70, 95% CI: 0.50-0.98, P=0.04) were related to a decreased risk in the presence of appetite loss (Table 2)”. (Page 14 Lines 2-7)

There is a typing mistake in the abstract:

“were related to decrease risk in the presence of appetite loss” instead of: “were related to decreased risk in the presence of appetite loss

Response: 

We have corrected the typing mistake. We appreciate your carful assessment.

---

## [Decision Letter · Decision Letter 3]

7 Apr 2022

Appetite Loss at Discharge from Acute Decompensated Heart Failure: Observation from KCHF registry

PONE-D-21-32941R3

Dear Dr. Kato,

We’re pleased to inform you that your manuscript has been judged scientifically suitable for publication and will be formally accepted for publication once it meets all outstanding technical requirements.

Kind regards,

Vincenzo Lionetti, M.D., PhD

Academic Editor

PLOS ONE

Additional Editor Comments (optional):

Reviewers' comments:

Reviewer's Responses to Questions

**Comments to the Author**

1. If the authors have adequately addressed your comments raised in a previous round of review and you feel that this manuscript is now acceptable for publication, you may indicate that here to bypass the “Comments to the Author” section, enter your conflict of interest statement in the “Confidential to Editor” section, and submit your "Accept" recommendation.

Reviewer #1: All comments have been addressed

2. Is the manuscript technically sound, and do the data support the conclusions?

Reviewer #1: Yes

3. Has the statistical analysis been performed appropriately and rigorously? 

Reviewer #1: Yes

4. Have the authors made all data underlying the findings in their manuscript fully available?

Reviewer #1: Yes

5. Is the manuscript presented in an intelligible fashion and written in standard English?

Reviewer #1: Yes

6. Review Comments to the Author

Reviewer #1: The authors have adequately addressed all comments raised in a previous round of review. I have no further comments or questions.

Congratulations to Authors on their work!

7. PLOS authors have the option to publish the peer review history of their article (what does this mean?). If published, this will include your full peer review and any attached files.

Reviewer #1: **Yes: **Anna Drohomirecka

---

## [Editor Report · Acceptance letter]

28 Apr 2022

PONE-D-21-32941R3 

Appetite Loss at Discharge from Acute Decompensated Heart Failure: Observation from KCHF registry 

Dear Dr. Kato:

I'm pleased to inform you that your manuscript has been deemed suitable for publication in PLOS ONE. Congratulations! Your manuscript is now with our production department. 

Kind regards, 

on behalf of

Prof. Vincenzo Lionetti 

Academic Editor

PLOS ONE